# Photo-switchable tweezers illuminate pore-opening motions of an ATP-gated P2X ion channel

Chloé Habermacher[1,2], Adeline Martz[1,2], Nicolas Calimet[3], Damien Lemoine[1,2], Laurie Peverini[1,2], Alexandre Specht[1,2], Marco Cecchini[3], Thomas Grutter[1,2]*

[1]Université de Strasbourg, Faculté de Pharmacie, Illkirch, France; [2]Centre National de la Recherche Scientifique, Laboratoire de Conception et Application de Molécules Bioactives, Unité Mixte de Recherche 7199, Équipe de Chimie et Neurobiologie Moléculaire, Illkirch, France; [3]ISIS, Unité Mixte de Recherche 7006, Laboratoire d'Ingénierie des Fonctions Moléculaires, Strasbourg, France

**Abstract** P2X receptors function by opening a transmembrane pore in response to extracellular ATP. Recent crystal structures solved in apo and ATP-bound states revealed molecular motions of the extracellular domain following agonist binding. However, the mechanism of pore opening still remains controversial. Here we use photo-switchable cross-linkers as 'molecular tweezers' to monitor a series of inter-residue distances in the transmembrane domain of the P2X2 receptor during activation. These experimentally based structural constraints combined with computational studies provide high-resolution models of the channel in the open and closed states. We show that the extent of the outer pore expansion is significantly reduced compared to the ATP-bound structure. Our data further reveal that the inner and outer ends of adjacent pore-lining helices come closer during opening, likely through a hinge-bending motion. These results provide new insight into the gating mechanism of P2X receptors and establish a versatile strategy applicable to other membrane proteins.

*For correspondence: grutter@unistra.fr

## Introduction

The family of P2X receptors encompasses seven subtypes in mammals (termed P2X1-7) that are widely expressed in many cells, including neurons (*Khakh and North, 2006*). These receptors are trimeric ion channels that switch rapidly between closed and opened conformations in response to extracellular ATP (*Browne et al., 2010*; *Jiang et al., 2013*). Once opened, a flow of ions (sodium, potassium and calcium) transit through the transmembrane (TM) pore and initiates signal transduction. Depending on the P2X subtype, gating is followed by desensitization, a temporary inactivation that terminates the ion flow. ATP-gated P2X receptors are also involved in a wide range of pathological disorders, including chronic and inflammatory pain (*Khakh and North, 2006*; *Abbracchio et al., 2009*). A detailed understanding of the molecular mechanisms underlying the gating process is thus of fundamental importance and may open new therapeutic avenues.

Recent X-ray structures of the zebrafish P2X4 (zfP2X4) receptor in its apo and ATP-bound states have provided a molecular understanding of how ATP binding triggers channel opening (*Kawate et al., 2009*; *Hattori and Gouaux, 2012*). There are three interfacial ATP-binding pockets in the extracellular domain (ECD) that protrude ~40 Å outside of the membrane. The transmembrane domain (TMD) comprises six TM spanning α-helices, two from each subunit termed TM1 and TM2, which are arranged around the threefold axis of symmetry, with TM1 positioned peripheral to TM2. The pore-lining TM2 helices are steeply angled nearly 45° from the membrane plane and

**eLife digest** Protein receptors in the cell membrane play an important role transmitting signals from outside to inside the cell. Members of the P2X family of receptors are ion channels that form pores through the membrane. When a molecule of ATP binds to the external region of the receptor, it activates it and causes the receptor to change from a closed to an open shape. Once opened, ions flow through the channel's pore and trigger a response inside the cell. P2X receptors are found on most animal cells (including nerve cells) and are involved in both normal cellular activity and processes linked to disease, including inflammation and chronic pain.

The P2X receptor has three parts or subunits, and each contributes to the channel's pore. Recent research using a technique called X-ray crystallography has revealed how ATP binding causes shape changes in the external region of the receptor. But these three-dimensional structures did not reveal details of how the subunits move to open or close the channel's pore.

Habermacher et al. have now added light-sensitive linkers onto the P2X receptor in a way that meant that different colors of light could be used to force parts of the receptor to come closer together or move apart. This allowed the pore to be opened and closed in response to changes in light. Habermacher et al. then studied the behavior of these modified receptors within a natural membrane and found that the light stimulated movements were similar to those seen with ATP. When the behavior of the receptor and light-sensitive linkers was studied using computer simulations, it led to new models of the P2X pore in the open and closed state. In these models, the open channel was more tightly packed than in the previous structure and an unexpected hinge-bending movement was seen to accompany the opening of the channel. It is hoped that this new approach will also be useful for probing how other membrane proteins change their shape when activated.

they form, in the apo state, a gate in the middle of the membrane that is thought to control the flux of ions. For this reason, the apo form is believed to represent a resting, closed state of the receptor (*Kawate et al., 2009*).

Structural and functional work has suggested that binding of ATP induces closure of the three interfacial pockets in the ECD that is accompanied by a rearrangement of the subunit–subunit interfaces (*Jiang et al., 2003*; *2010*; *2012*; *Nagaya et al., 2005*; *Marquez-Klaka et al., 2007*; *Du et al., 2012*; *Hattori and Gouaux, 2012*; *Lorinczi et al., 2012*; *Roberts et al., 2012*; *Hausmann et al., 2013*; *Huang et al., 2014*; *Stelmashenko et al., 2014*; *Zhao et al., 2014*). As a result, the lower region of the ECD undergoes a flexing motion that pulls apart the outer ends of the six TM helices (*Hattori and Gouaux, 2012*). This lateral displacement, in turn, enlarges through an iris-like opening the narrowest part of the channel, creating a wide entryway of 7 Å in diameter, which allows ions to flow through the open pore (*Hattori and Gouaux, 2012*).

The mechanism of gating based on the crystal structures is largely consistent with previous functional and modeling data obtained on the ECD (*Jiang et al., 2010*; *2012*; *Du et al., 2012*; *Lorinczi et al., 2012*; *Roberts et al., 2012*; *Hausmann et al., 2013*; *Huang et al., 2014*; *Stelmashenko et al., 2014*; *Zhao et al., 2014*). However, there are areas of discordance between the X-ray structures and the available data at the level of the TM pore. Although the location of the gate (*Samways et al., 2014*), the relative position and gating motion of TM1 and TM2 within the individual subunits (*Li et al., 2008*; *Heymann et al., 2013*) and the movement of the outer ends of the TM helices (*Li et al., 2008*; *2010*; *Kracun et al., 2010*; *Heymann et al., 2013*; *Browne et al., 2014*) inferred from experimental data are in qualitative agreement with the crystal structures, there are reasons to question whether the ATP-bound structure provides an accurate blueprint of a native open-channel pore. First, the proposed mechanism of gating implies a great radial outward movement of the TM helices away from the threefold axis, which creates large 'crevices' between the TM helices of adjacent subunits. As a result, the TMD appears loosely packed. Although endogenous lipids have been suggested to occupy these gaps (*Hattori and Gouaux, 2012*), recent modeling supported by experimental data has suggested that these crevices are not present in membrane-embedded receptors (*Heymann et al., 2013*). Second, metal bridging experiments suggest that the inner portion of TM2 helices narrows as the channel opens (*Kracun et al., 2010*; *Li et al., 2010*), a

feature that is not visible from the ATP-bound structure (*Hattori and Gouaux, 2012*). Third, to obtain diffracting crystals, proteins were extensively truncated at both the amino and carboxy termini and solubilized with detergents (*Hattori and Gouaux, 2012*). Although the truncated receptor was functional, the lack of the intracellular domains, which critically control the function of the receptor (*Chaumont et al., 2004*; *Allsopp and Evans, 2011*; *Robinson and Murrell-Lagnado, 2013*), may distort somewhat the structure of the pore, raising the possibility that the X-ray structure may represent a non-native, open-channel state. Hence, additional data are needed to understand the gating mechanism of a full-length, membrane-embedded P2X receptor.

In this study, we present a new chemical-based method to explore the gating motion of the rat P2X2 (rP2X2) receptor by cross-linking engineered cysteine residues in the TMD with photo-switchable azobenzene tweezers, for which the end-to-end distance can be controlled by light. A set of structural constraints combined with Molecular Dynamics (MD) provided high-resolution models of the channel both in the open and closed states in their physiological environment. These results provide unprecedented insight into the pore-opening motions of this major class of ligand-gated ion channels.

## Results

### Designing the photo-switchable tweezers strategy

To collect structural constraints related to channel gating, we synthesized 4,4′-bis(maleimido-glycine) azobenzene (MAM), a semi-rigid photo-switchable azobenzene cross-linker carrying two sulfhydryl-reactive maleimides known to cross-link pairs of engineered cysteine residues (*Figure 1A* and *Figure 1—figure supplement 1A*). In solution, MAM rapidly isomerizes from *trans* to *cis* configuration at 365 nm and reversibly switches back to the *trans* isomer at 525 nm (*Figure 1—figure supplement 1B*) or slowly by thermal relaxation ($\tau = 388 \pm 28$ min, n = 4 in DMSO). As a consequence, the end-to-end distance of MAM can be changed by light from $21.7 \pm 2.0$ Å in the *trans* state to $16.0 \pm 4.6$ Å (n = 100000 conformers) in the *cis* state, as measured from the distribution of the S–S distance in explicit-water MD simulations (*Table 1* and *Figure 1—figure supplement 1C*, thin dashed lines); see Experimental Procedures for computational details.

### Horizontal cross-linking provides a direct measurement of the outward expansion of the TM2 helices during activation

Guided by P2X2 homology models (*Lemoine et al., 2013*) built from zfP2X4 X-ray structures of the apo (PDB code: 4DW0) (*Hattori and Gouaux, 2012*) and the ATP-bound states (4DW1) (*Hattori and Gouaux, 2012*), we selected pairs of residues within the TMD to be mutated into cysteine. We chose residues from the extracellular apex of the TM helices for which the interatomic distances of the Cβ-atoms of at least one of the X-ray structures closely matched the end-to-end distance of MAM (*Figure 1—source data 1*). Furthermore, to limit all possible combinations, only single cysteine mutations per subunit were designed, and, therefore, only one MAM photo-linker is expected to cross-link two adjacent subunits within a receptor. As pairwise positions are considered, horizontal cross-linking relative to the membrane plane is expected. With these criteria, we identified 10 residues in TM1 and TM2 (*Figure 1B*) and substituted each of them with a cysteine into the P2X2-3T background, a functional receptor in which the three native cysteines were mutated to threonine (*Li et al., 2008*), and expressed the single mutants in HEK cells. All cysteine mutants responded robustly to ATP, as assayed by whole-cell patch-clamp electrophysiology, and displayed half-maximal effective concentrations (EC$_{50}$) of ATP that were similar to those determined previously (*Figure 1—source data 2*) (*Li et al., 2008*; *Lemoine et al., 2013*).

We screened each mutant by recording currents in response to light irradiation from cells treated with MAM for 20 min in the dark. Because MAM is expected to open the channel in *trans*, cells were illuminated at 365 nm prior to 525 nm. No light-gated currents were observed for the P2X2-3T, Q52C, and P329C mutants (*Figure 1C, D* and *Figure 1—figure supplement 2A*). However, for all remaining mutants, channels reversibly opened at 525 nm light and closed at 365 nm light, except for N333C, which responded weakly in the opposite sense to these wavelengths (*Figure 1C, D* and *Figure 1—figure supplement 2A*).

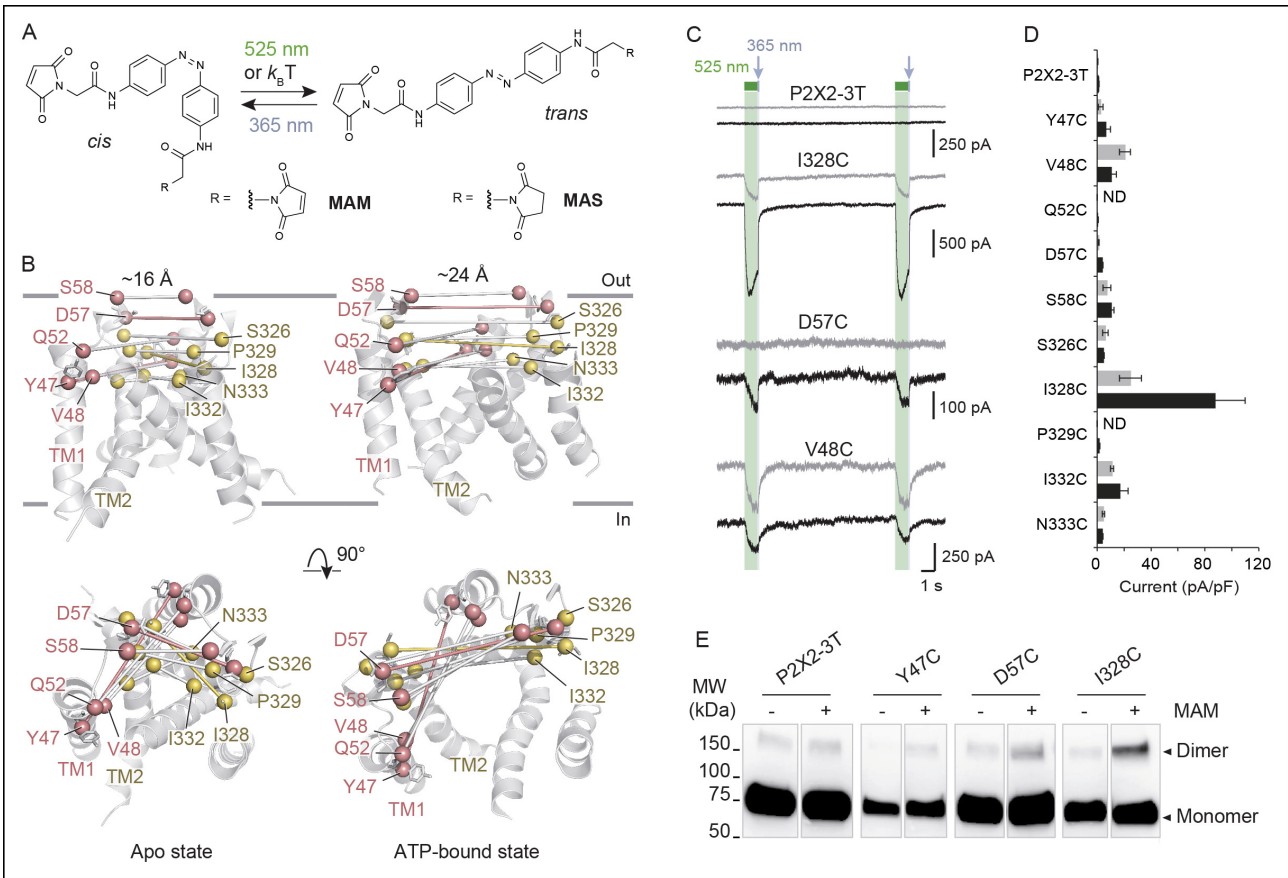

**Figure 1.** Lateral expansion between TM1 and TM2 helices drives channel opening. (A) Chemical structures of MAM and MAS in the *cis* and *trans* states. (B) Cartoon representation of the TMD of a P2X2 homology model viewed parallel (upper) and perpendicular (lower) to the membrane plane in an apo (left) and ATP-bound state (right). Cβ atoms of residues selected for cysteine substitutions are shown as red and yellow spheres in TM1 and TM2 helices, respectively. Indicated values are the average distances separating pairwise β-atoms from two adjacent subunits (grey bridges). Highlighted bridges indicate actual MAM cross-linking. (C) Whole-cell currents recorded during and after illumination at 525 nm (green bars, 1 s) and 365 nm (violet arrows, 80 ms) in HEK cells expressing the P2X2-3T receptor or the indicated cysteine-substituted mutants after treatment with MAM (black traces) or MAS (gray traces). Just before recordings, cells were irradiated for 85 ms with a light pulse of 365 nm. (D) Screening for all constructs showing light-gated currents following MAM (filled bars) or MAS (gray bars) treatment. All light-gated mutants were activated at 525 nm and inactivated at 365 nm, except for N333C, which responded in the opposite sense to these wavelengths. ND stands for not determined (n = 4–5 cells; mean ± s.e.m.). (E) Western blot analysis of cell-surface cross-linking of the indicated P2X2-3T constructs expressed in TSA-201 cells after treatment (+) or without treatment (-) with MAM. Monomer and dimer are indicated. Uncut gel image is shown in *Figure 1—figure supplement 2B*. MAM: 4,4′-bis(maleimido-glycine)azobenzene; MAS: 4-(maleimido-glycine)-4'-(succimido-glycine)azobenzene; MW: Molecular weight; TMD: Transmembrane domain.

The following source data and figure supplements are available for figure 1:

**Source data 1.** Interatomic distances between pairwise residues.

**Source data 2.** Estimated $EC_{50}$ and Hill coefficients for ATP activation.

**Figure supplement 1.** Chemical synthesis and physico-chemical properties of azobenzene derivatives.

**Figure supplement 2.** Horizontal screening confirms an outward expansion of the TM helices.

**Figure supplement 3.** Characterization of currents induced by the isomerization of azobenzene compounds attached at the I328C or Y47C mutant.

**Figure supplement 4.** [1]H and [13]C NMR of MAM (1).

**Figure supplement 5.** [1]H and [13]C NMR of 2.

*Figure 1 continued*

**Figure supplement 6.** $^1$H and $^{13}$C NMR of MAS (3).

To confirm that these currents originated from successful cross-linking of two adjacent cysteines, we synthesized another azobenzene derivative, named MAS (4-(maleimido-glycine)-4'-(succimido-glycine)azobenzene), in which one of the two maleimides was replaced by the isosteric, sulfhydryl non-reactive succinimide (*Figure 1A* and *Figure 1—figure supplement 1A*). In these conditions, MAS is expected to react with a single cysteine without cross-linking. Control experiments with MAS revealed that the majority of the light-gated currents were indistinguishable from those originating from MAM treatment (*Figure 1C, D* and *Figure 1—figure supplement 2A*), suggesting that in these experiments MAM actually does not cross-link adjacent subunits. However, for three mutants (Y47C, D57C, and I328C) there was clear evidence for MAM cross-linking. For the D57C mutant, there was no light-gated current following MAS treatment, whereas the response of the Y47C and I328C mutants to MAM exhibited clear differences relative to MAS in current amplitudes (*Figure 1D*), on-rate kinetics and stability in the dark (*Figure 1—figure supplement 3A–D*). Moreover, biochemical experiments clearly showed that the I328C-mutated protein, and not P2X2-3T, migrated on sodium dodecyl sulphate polyacrylamide gel electrophoresis (SDS–PAGE) at the position expected for dimeric cross-linked subunits only after MAM treatment (*Figure 1E*). Similar results were also obtained for the Y47C and D57C mutants (*Figure 1E*), although the extent of cross-linking was lower than that of I328C. The reason for the relatively low extent of cross-linking is unclear, but the currents elicited by light were also consistently small for these mutants compared to the I328C mutant, suggesting that the kinetics of cross-linking are not complete. Alternatively, a fraction of the cell-surface mutant receptors might not be reactive to MAM fusion (possibly due to protein misfolding), thus decreasing the cross-linking efficiency. Finally, we cannot rule out the possibility that the dimer band transfers less efficiently in Western blotting. Overall, because the Y47, D57, and I328 residues are located at the outer ends of the TMD, these cross-linking data provide direct evidence that a lateral outward motion of both TM1 and TM2 is involved in channel gating.

Quantification of the cross-linking results reveals that the average value from MD simulations of the end-to-end distance of the free *cis* isomer in solution closely matches the average Cβ–Cβ distance between I336 residues (equivalent to I328 in rP2X2) in the X-ray structure of the closed state zfP2X4 (*Table 1*). By contrast, the average value of the *trans* isomer in solution is 6 Å shorter than the Cβ–Cβ distance measured between the same pairs of residues in the structure of the open state. Thus, the crystallographic structure of the channel in the open state appears to be inconsistent with the present cross-linking experiments.

An interesting behavior of the azobenzene fused horizontally between two I328C mutant subunits was revealed when channels previously turned off by a brief illumination at 365 nm were still able to respond to the same wavelength of light, although currents were smaller than those elicited at 525 nm light (*Figure 1—figure supplement 3E*). As these results were only observed with MAM, they suggest that the channel acts as a 'lever' to maintain in its resting state the cross-linked azobenzene in the high-energy *cis* state. Consistent with photostationary states, we propose that a small but significant proportion of *trans* state is formed at 365 nm besides the large proportion of *cis* state, causing the observed small inward currents.

## Light-gated motions as mimicry of ATP-gated motions

To evaluate the physiological relevance of the light-driven motions, we determined key biophysical features of the light-gated channels and compared them to those of channels activated by ATP. We focused on the I328C mutant because it gave the most robust currents. First, we found that the rate of activation by light (time constant τ = 131 ± 9 ms, n = 8, *Figure 2A*) closely resembled that induced by a saturating concentration of ATP for the wild-type (WT) P2X2 receptor (300 μM; τ = 128 ± 11 ms, n = 4) (*Trujillo et al., 2006*). This result suggests that light and ATP open the channel with similar kinetics, although light source intensity and conditions of ATP application were not optimal. Second, increasing time treatment with MAM from 5 to 40 min increased the relative amount of light-gated currents relative to maximal ATP-gated currents from 20 ± 7 to 60 ± 5% (n = 7–9), demonstrating that light promotes gating motions that are nearly as efficient as those induced by ATP

**Table 1.** Comparison of the Cβ–Cβ distances (in Å) in the crystal structures of the closed and open states and after MD relaxation with fused MAM, along with the end-to-end distances for the free MAM in solution[a].

| State/isomer | Horizontal cross-linking | | Vertical cross-linking | |
| --- | --- | --- | --- | --- |
| | closed/cis | open/trans | closed/trans | open/cis |
| X-Ray | 16.1 | 27.7 | 20.8 | 23.2 |
| Free MAM MD | 16.0 ± 4.6 | 21.7 ± 2.0 | 21.7 ± 2.0 | 16.0 ± 4.6 |
| △ from X-Ray | -0.1[b] | -6.0 | 0.9 | -7.2 |
| Fused MAM MD | 17.1 ± 0.5 | 24.7 ± 0.6 | 18.9 ± 0.9 | 17.2 ± 1.0 |
| △ from X-Ray | +1.0 | -3.0 | -1.9 | -6.0 |

[a]The distances are measured between the Cβ atoms of the native or cysteine-cross-linked residues, that is, I336(C) for horizontal cross-linking and I336(C)/N353(C) for vertical cross-linking, or between the S atoms for the free MAM. [b]Note that in this case, Δ gives only an estimate of the difference because it was determined from the S-S distances of free MAM and Cβ–Cβ distances of the X-ray data.

MD: Molecular dynamics; MAM: 4,4′-bis(maleimido-glycine)azobenzene.

(*Figure 2—figure supplement 1A-C*). Third, light-gated currents rapidly inactivated in the dark, and currents were recovered from inactivation by switching back to the *cis*-isomer (*Figure 2—figure supplement 2*), two processes that are reminiscent of P2X desensitization and resensitization, respectively. Fourth, channels opened by light at 525 nm increased by twofold the maximal ATP response without changing ATP sensitivity relative to control. Given that ATP is a partial agonist on the rP2X2 receptor (*Ding and Sachs, 1999*), this suggests that MAM-induced motions are functionally linked to ATP function (*Figure 2—figure supplement 1D, E* and *Figure 1—source data 2*). Fifth, we found that light-gated channels remained selective to cations (*Figure 2—source data 1*) and showed calcium permeability that was somewhat lower (by 1.6-fold) than that of the ATP-gated P2X2-3T (*Figure 2B* and *Figure 2—source data 2*) (*Lemoine et al., 2013*). In addition, as P2X2 receptors are known to undergo 'pore dilation' during prolonged ATP application (*Khakh et al., 1999*; *Virginio et al., 1999*; *Rokic and Stojilkovic, 2013*), which is defined by a progressive increase in permeability to large organic cations such as *N*-methyl-d-glucamine (NMDG), we tested the ability of NMDG to permeate light-gated channels. No NMDG current was detected after 525 nm light switching (*Figure 2—source data 3*), suggesting that the open state reached by MAM fused horizontally between two I328C mutant subunits in the *trans* configuration is not trapped in a dilated, open-channel state (*Figure 2B*). Last, to determine the unitary conductance of the light-gated receptors, we carried out single-channel recordings in the outside-out configuration (*Figure 2C*). Control experiments showed that the main conductance state (O) of channels opened by ATP in patches excised from cells expressing the P2X2-3T channel was somewhat higher than that previously determined for the WT P2X2 receptor (*Jiang et al., 2011*; *Rothwell et al., 2014*). This increased ATP unitary current enabled us to detect an additional low-conductance substate (S), a feature that has already been reported to occur occasionally for the WT P2X2 receptor (*Figure 2C* and *Figure 2—source data 4*) (*Ding and Sachs, 1999*). In agreement with the macroscopic recordings, illumination at 525 nm of patches excised from MAM-treated cells expressing the I328C mutant induced a peak current that rapidly declined to a steady-state level of activity, where individual openings and closings can be detected (*Figure 2C*). Compared to the P2X2-3T receptor, currents opened and closed much more frequently and were highly flickery. Due to this flickery behavior, accurate determination of the open and shut times could not be made. This finding is in agreement with a recent report showing that unitary currents of the I328C channels were also flickery and that receptors exhibit a degree of activity in the absence of exogenously applied ATP (*Rothwell et al., 2014*). However, analysis of the all-points histograms showed that channels appeared to open to multiple conducting levels, as observed for the ATP-gated P2X2-3T receptor. These conductance states (full (O) and low-conductance substates (S)) were comparable, albeit lower (~35%) than those of the P2X2-3T receptor (*Figure 2—source data 4*), suggesting that the open states reached by MAM under light illumination are not fundamentally dissimilar to those naturally populated by ATP.

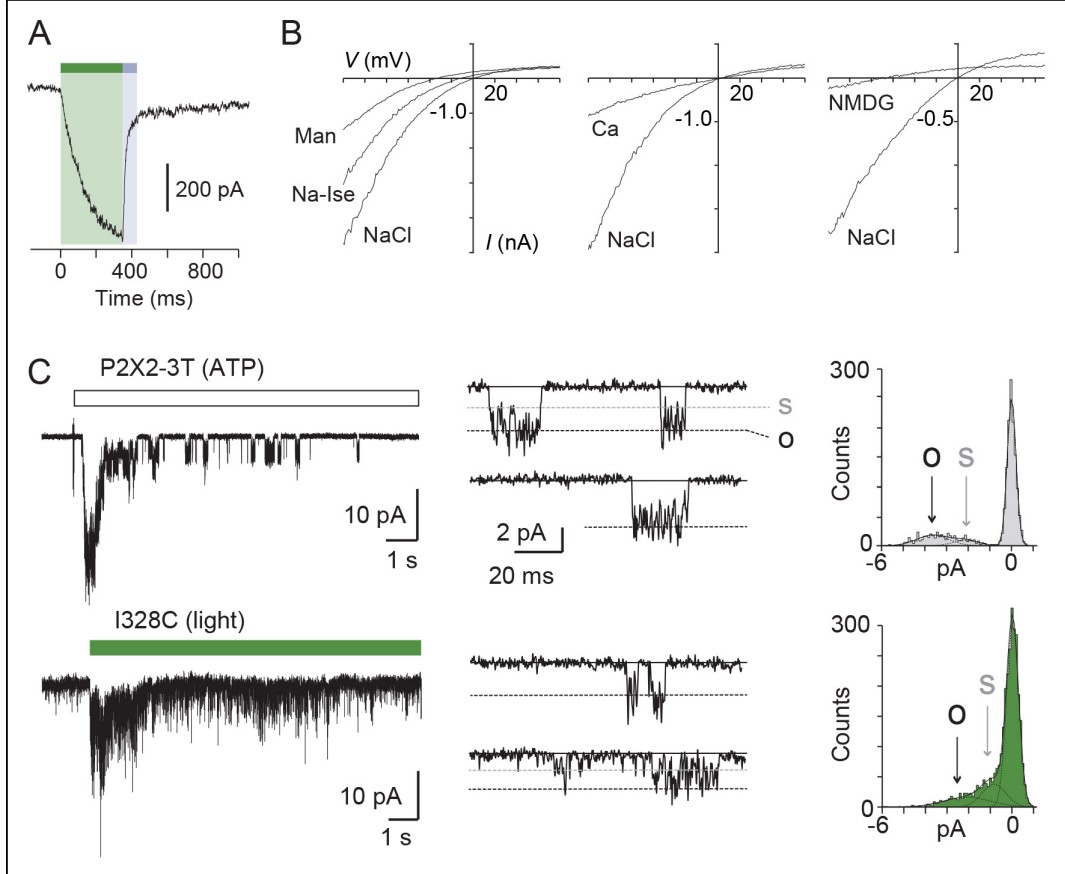

**Figure 2.** Light-driven open states are similar to those induced by ATP in the I328C mutant. (A) Optimized illumination times at 525 nm (green bar, 350 ms, 4.1 mW/mm$^2$) and 365 nm (violet bar, 80 ms, 8.1 mW/mm$^2$) of I328C mutant treated with MAM to observe maximal opening and closing. (B) Current-voltage curves recorded in different extracellular solutions (Man, mannitol; Na-Ise, sodium isethionate; Ca, calcium; NaCl, symmetrical NaCl external solution; NMDG, N-methyl-D-glucamine). Shown are light-gated currents obtained after subtracting peak photocurrents recorded at 525 nm light to those obtained in the dark after switching to 365 nm light. (C) Left, single-channel currents recorded from outside-out patches at -120 mV in response to ATP for the P2X2-3T (10 μM, upper panel) or to 525 nm illumination for I328C mutant treated with MAM (4.1 mW/mm$^2$, lower panel). In these conditions, both ATP- and light-gated currents correspond to ~30% of a maximal ATP response. Middle, unitary currents shown on an expanded scale. Full (O) and sublevel (S) openings are indicated by dashed black and gray lines, respectively. Black lines indicate closed channels. Right, corresponding all-points histograms, fitted to the sum of three Gaussians. Full and sublevel openings are also indicated.

The following source data and figure supplements are available for figure 2:

**Source data 1.** Relative ion permeability for chloride.
**Source data 2.** Relative ion permeability for calcium.
**Source data 3.** Relative ion permeability for NMDG.
**Source data 4.** Single-channel properties of light-gated and ATP-gated receptors.
**Figure supplement 1.** Kinetics of MAM labeling and effect of light on ATP currents in cells expressing the I328C mutant.
**Figure supplement 2.** Exploration of desensitization and resensitization of the I328C mutant treated with MAM.

## Vertical cross-linking indicates shortening of the distance separating the inner and outer ends of adjacent TM2 helices during activation

We next investigated movements of the pore that occur during activation along an axis perpendicular to the membrane plane (i.e. vertical motion). We paired the mutation I328C located at the

extracellular apex of one TM2 with other cysteine substitutions located at the intracellular side of another TM2 (from V343 to W350; *Figure 3A*), thus designing double-cysteine mutants. To distinguish vertical MAM cross-linking from either horizontal cross-linking or MAM reactions involving only one I328C residue (i.e. such as with MAS), light-gated currents were systematically compared with those originating from the I328C mutant alone. All single and double mutants gave robust ATP currents, except for I328C/D349C, which was not functional and D349C, which gave unstable currents (*Figure 1—source data 2*). These latter two mutants were not further analyzed. Following MAM treatment, none of the single internal pore mutants responded to light (*Figure 3B* and *Figure 3—figure supplement 1A*). In contrast, all the functional double mutants were light sensitive, although for most of them activation at 525 nm was very similar to that of the I328C mutant, suggesting that cross-linking occurs horizontally between two adjacent I328C-mutant subunits (*Figure 3B*).

The situation was reversed for the I328C/S345C mutant (and to a lesser extent I328C/F346C), whose channel opened in the *cis* configuration and closed in the *trans* configuration of the photolinker (*Figure 3B*). In control experiments, the same mutant treated by MAS responded to the opposite senses of these wavelengths (*Figure 3—figure supplement 1B*), and no light-gated current was recorded when the serine mutant I328S/S345C was treated with MAM (*Figure 3B*). Biochemical experiments firmly demonstrated inter-subunit vertical cross-linking as visualized by the expected presence of dimeric and trimeric cross-linked species (*Figure 3C*).

Next, we show that light-gated channels exhibit biophysical features of WT receptors activated by ATP. First, light-gated motions were nearly as effective as those induced by ATP, as evidenced by the fact that light activated channels at $60 \pm 7\%$ (n = 13 cells, preincubation time of 20 min) of the maximal whole-cell ATP response. This value was higher than that obtained from I328C measured at the same preincubation time ($31 \pm 4\%$), suggesting that rates of vertical cross-linking between I328C and S345C are higher than those observed horizontally involving two I328C-mutated subunits. Furthermore, in contrast to the single I328C mutant, light-gated currents were stable in the dark for I328C/S345C double mutant, suggesting that these light-driven motions did not induce apparent desensitization of the receptor (*Figure 3—figure supplement 2A*). Second, engineered I328C/S345C channels opened by *cis*-MAM were still largely selective for sodium over chloride ions (*Figure 2—source data 1*), and displayed calcium permeability that was higher than that of the I328C mutant, but slightly lower than that of the P2X2-3T receptor (*Figure 3—figure supplement 2B* and *Figure 2—source data 2*). Third, a small but substantial NMDG permeability (around 15%) was recorded (*Figure 2—source data 3*), which suggests that *cis*-MAM traps a partially dilated state, revealing that the MAM photo-linker has the ability to trap different ion permeability states. Last, in outside-out patches, channels opened stepwise after 365 nm illumination to discrete levels of conductance with no apparent decline of activity, in agreement with macroscopic recordings (*Figure 3D*). Unitary currents of the highest conducting level ($O_1$ in *Figure 3E*) were higher than those of the I328C mutant cross-linked horizontally by MAM and resembled those of the P2X2-3T receptor activated by ATP, although the probability of occurrence was extremely low (*Figure 2—source data 4*). Two additional low-conductance substates ($S_1$ and $S_2$) were also detected in relatively abundant amounts that may correspond to a mixture of cross-linked species (*Figure 3E*). Because currents were also highly flickery, they were not analyzed further. Overall, these data indicate that shortening the distance between the inner and outer ends of adjacent TM2 helices drives channel opening, a motion that was not anticipated by X-ray structures.

Quantification of the cross-linking results showed that although the distance between the Cβ atoms of the vertical anchor points in the X-ray structure of the closed state zfP2X4 (equivalent to I328 and S345 in rP2X2) matches the average end-to-end distance of free MAM in *trans*, the same is not true in the structure of the open state, where the Cβ–Cβ distance is considerably larger (~7 Å) than the average end-to-end distance in free MAM in *cis* (*Table 1*). Consistent with the horizontal cross-linking results, these data support the conclusion that the X-ray structure of the channel in the ATP-bound zfP2X4 may represent a non-native, open-channel state.

## Light-induced decreasing of the distance between TM2 ends, on at least two subunit interfaces, increases ATP function

We next addressed the contribution of the vertical shortening of TM2 ends to the ATP response. We found that MAM cross-linking increased the time constants (τ) of ATP washout by twofold from $0.580 \pm 0.092$ (n = 4, before treatment) to $1.062 \pm 0.090$ s (n = 7, after MAM treatment, *Figure 4A*).

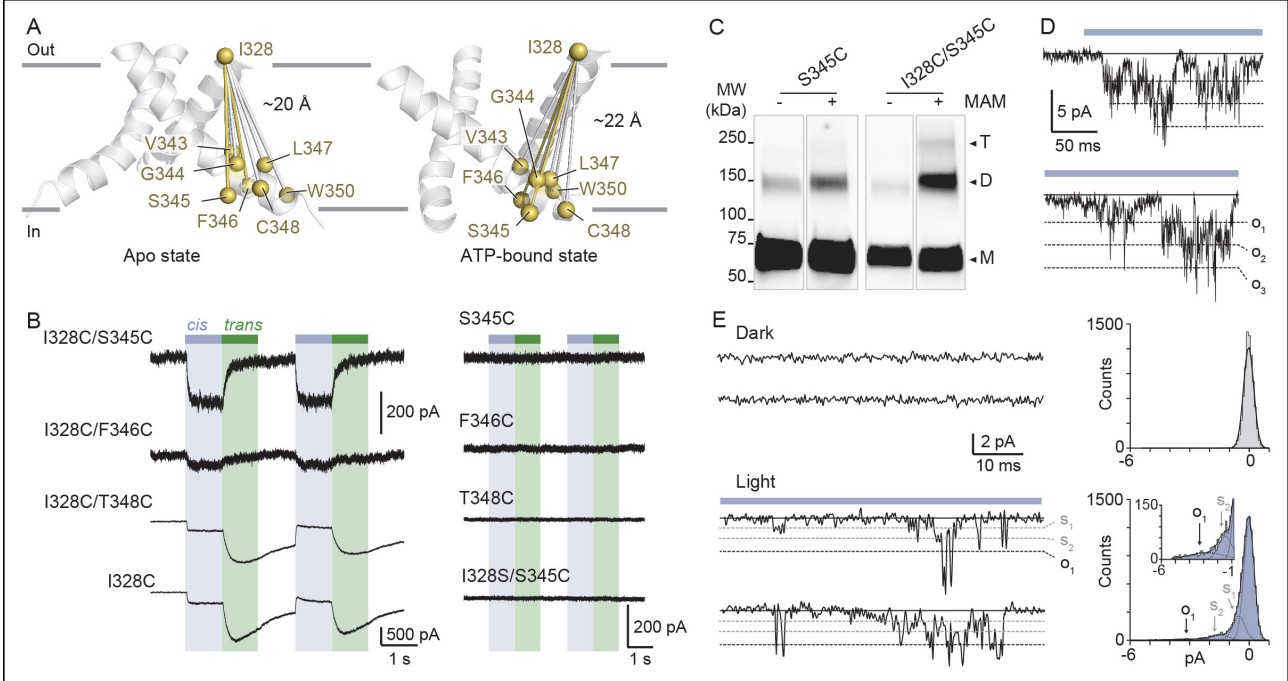

**Figure 3.** Shortening of the vertical distance separating adjacent TM2 ends drives channel openings. (**A**) Side views of TM2 helices of a P2X2 homology model in the apo state (left) and ATP-bound state (right). The β-atoms of residues selected for cysteine substitutions are shown as yellow spheres. Indicated values are the average distances separating pairwise β-atoms from two adjacent TM2 helices (grey bridges). Highlighted bridges between residues indicate actual MAM cross-linking. For clarity, TM1 helices are omitted. (**B**) Whole-cell currents evoked by light at 365 nm (*cis*) or 525 nm (*trans*) in HEK cells expressing the indicated cysteine-substituted mutants treated with MAM. (**C**) Western blot analysis of cell-surface cross-linking of the indicated mutated subunits expressed in TSA-201 cells after treatment (+) or without treatment (-) with MAM. Monomer (M), dimer (D), and trimer (T) are indicated. Uncut gel image is shown in *Figure 3—figure supplement 1C*. MW, Molecular weight. (**D**) Single-channel currents recorded from an outside-out patch expressing the I328C/S345C mutant at -120 mV in response to 365 nm illumination. Three simultaneous openings (O) indicated by dashed black lines were detected. Black lines indicate closed channels. (**E**) Unitary currents (left) and corresponding all-points histograms (right) recorded before (upper) and during 365 nm illumination (lower) from the same patch expressing the I328C/S345C mutant. Sublevel openings (S1 and S2) are indicated by dotted gray lines. Inset shows expanded scale. All-points histograms were fitted to one Gaussian (upper) or to the sum of four Gaussians (lower).

The following figure supplements are available for figure 3:

**Figure supplement 1.** Vertical screening identifies a shortening of the distance separating adjacent TM2 ends during activation.
**Figure supplement 2.** Biophysical properties of the I328C/S345C mutant.
**Figure supplement 3.** Concatenated P2X2-3T receptors are gated by UV light with only two cross-linked MAM.

Noteworthy is that toggling bound azobenzene in the *cis* state both increased maximal current by 1.4-fold (*Figure 4B*) and ATP sensitivity by fourfold (n = 4 cells) (*Figure 4C* and *Figure 1—source data 2*) compared to currents measured in the *trans* state. This demonstrates that the light-driven motions of the TM pore are cooperatively transmitted to the distant, extracellular ATP-binding sites.

To identify the minimal number of bound MAM that is necessary to open the pore, we designed concatenated cDNAs that encoded three joint P2X2-3T subunits, which had the I328C and/or S345C mutation(s) in one, two, or three subunits (*Figure 3—figure supplement 3A*). There are potential caveats concerning the use of concatemers (*Stelmashenko et al., 2012*), but we provide evidence that currents recorded after expression of the different constructs result from channels formed by the intact concatenated proteins. All tested concatemers responded robustly to ATP, and no apparent protein breakdown was detected by SDS–PAGE (*Figure 3—figure supplement 3B and 3D*). Concatemers that allow MAM cross-linking at one of the three interfaces (denoted OO/CO/OC or OC/OO/CO, where C and O stands for cysteine mutation and WT residue, respectively) or within

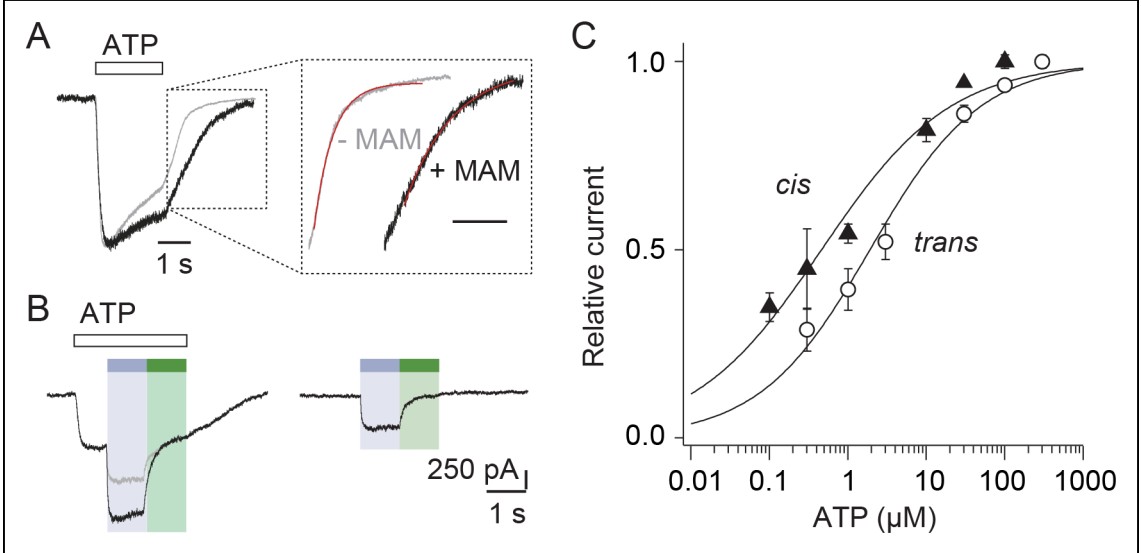

**Figure 4.** Vertical motions induced by *cis*-MAM at the I328C/S345C mutant increase ATP function. (**A**) Normalized whole-cell currents evoked by a saturating concentration of ATP (100 µM) recorded before (gray trace) and after (black trace) treatment with MAM. Inset highlights part of the currents upon ATP washout, fitted to single exponential decay functions (red traces). (**B**) Whole-cell light-gated currents recorded from the same cell in the presence (left) or absence (right) of ATP (100 µM). Gray trace indicates the predicted current if ATP-gated and light-gated currents at 365 nm were additive. (**C**) Concentration–response relationships for ATP at 525 nm (open circle, *trans* state) or in the dark immediately after illumination at 365 nm (filled triangle, *cis* state) (n = 4–8 cells; mean ± s.e.m.). Currents were normalized to 100 µM ATP at 365 nm and 300 µM ATP at 525 nm. The Hill equation was fit to the data.

one subunit (intra helix cross-linking; OO/OO/CC) showed no light-gated currents (*Figure 3—figure supplement 3A–C*). However, a gradual increase of the ratio of the currents elicited by UV light to those gated by ATP was observed for concatemers that allow MAM cross-linking at two and three interfaces assuming a clockwise orientation of the subunits (OC/CO/CC and CC/CC/CC, *Figure 3—figure supplement 3A–C*), demonstrating that two MAM linkers are sufficient to gate the channel (the alternative counterclockwise orientation was inconsistent with the data). No or negligible light-gated currents were recorded in cells expressing concatemers harboring cysteine mutations in other combinations, as would be expected if there were alternative patterns of cross-linking (*Figure 3—figure supplement 3A–C*).

## Improved model of the open-channel state

To provide a structural interpretation of the light-gating experiments, we produced atomistic models of the closed and open states by explicitly incorporating MAM photo-linkers in the apo and the ATP-bound X-ray structures of the zfP2X4 receptor (*Hattori and Gouaux, 2012*). For this purpose, homologous cysteine mutations were introduced (I336C and N353C equivalent to rP2X2 I328C and S345C, respectively) and the mutants were fused with MAM photo-linkers in both *cis* and *trans* isoforms. The resulting constructs were then relaxed by short MD simulations in explicit water and membrane.

In both vertical and horizontal crosslinking simulations, the structural restraints imposed by MAM had little effect on the structure of the closed-channel state relative to the X-ray structure of apo zfP2X4 (*Figure 5A*). In sharp contrast, the MD relaxation of the open state with fused MAM produced a significant contraction of the TMD, as compared to the X-ray structure of the ATP-bound state with the anchor points distances shortening from 27.7 to 24.7 Å upon horizontal cross-linking (Cβ of I336C) and from 23.2 to 17.2 Å upon vertical cross-linking (Cβ of I336C and N353C); see *Figure 5B* and *Table 1*. Consistent with a recent modeling study (*Heymann et al., 2013*), this structural rearrangement results in the disappearance of the large crevices observed in the X-ray structure of the open state (*Figure 5D*). In addition, the cross-linking simulations of the open state produce a more compact structure of the inner TMD, with an open channel stabilized by a new interface between the pore-lining TM2 helices (note that for the simulations of the open state, the TM1-TM2

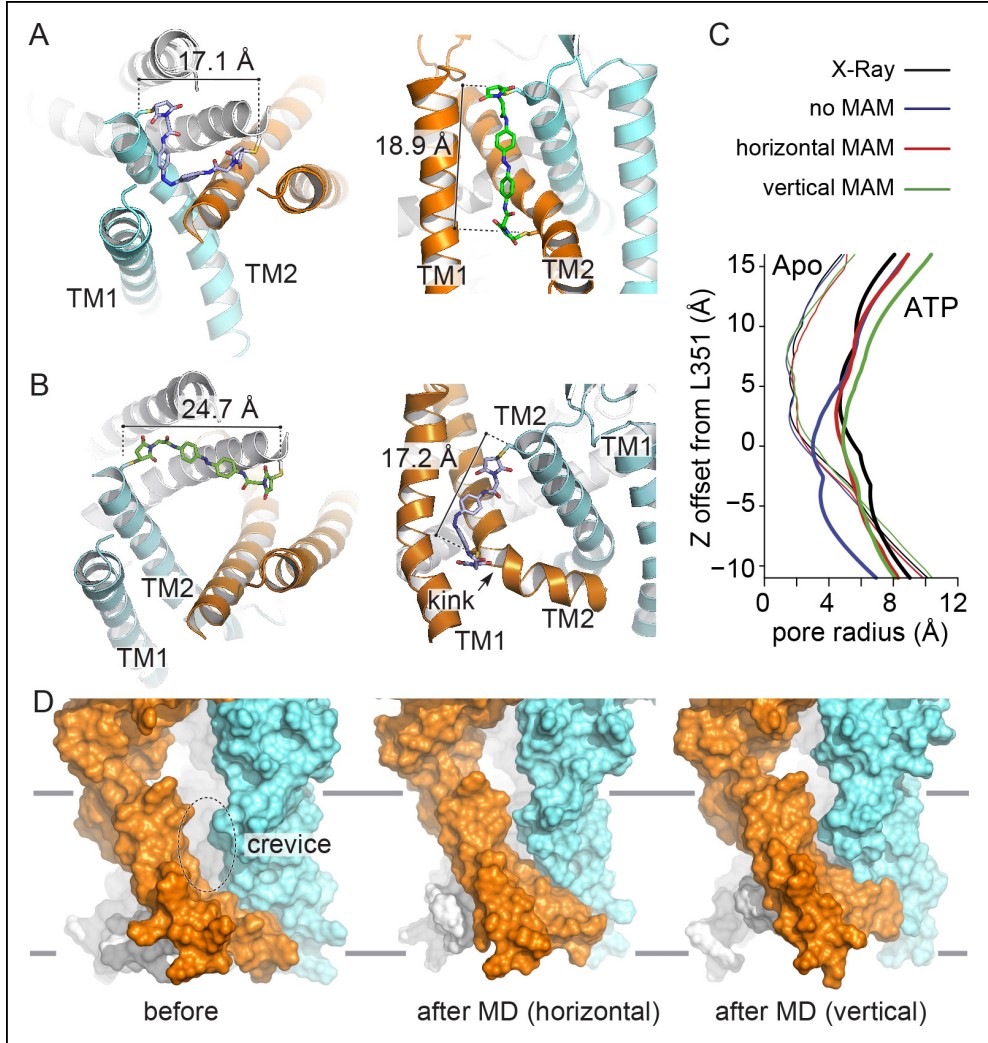

**Figure 5.** Molecular dynamics of zfP2X4 open-channel state cross-linked by MAM reduce inter-subunit interface in the TMD. (**A**) Cartoon representation of the TMD of zfP2X4 receptor simulated in the closed state after MD, in which MAM (in stick representation) is conjugated horizontally between two I336C (*cis* configuration, left) or vertically between I336C and N353C (*trans* configuration, right). For clarity, only one MAM is shown vertically. Distances separating Cβ atoms of engineered cysteines are 17.1 ± 0.5 Å (n = 800) and 18.9 ± 0.9 Å (n = 2400). (**B**) Same views of the TMD simulated in the open state. Distances separating Cβ atoms are 24.7 ± 0.6 Å (n = 800) and 17.2 ± 1.0 Å (n = 2400). TM1 and TM2 helices and the location of a kink in one of the three TM2 helices are also shown. (**C**) Transmembrane pore radius along the axis of the ion channel for the apo (thin lines) and ATP-bound (thick lines) states. The profiles were calculated considering the backbone atoms only (see Materials and methods) and were derived from the X-ray structures (black) or models obtained after MD relaxation computed with (red and green) or without MAM (blue) as indicated. In the absence of MAM, the open state rapidly closes in MD simulations near L351 (rP2X2: V343). (**D**) Lateral view of the channel displayed in surface representation before (left) and after MD following MAM attachment between two I336C (middle) or between I336C and N353C (right). MAM: 4,4′-bis(maleimido-glycine)azobenzene; MD: Molecular dynamics; TMD: Transmembrane domain.

The following figure supplement is available for figure 5:

**Figure supplement 1.** Comparison between the ATP-bound crystal structure and the new model of the open state.

interface was stabilized by incorporating structural information obtained from recent intra-subunit Cd$^{2+}$ bridging essays [*Heymann et al., 2013*]). This new interface involves the side chain of L351 (rP2X2: V343) on one subunit and G343 (A335) and/or that of A347 (T339) on the adjacent subunit. The restructuring of the TMD in simulation is driven by a significant conformational change of the upper region of the TM2 helices, which straighten, pack closer together around the region of L351 and lock the ion channel in an open-pore conformation (*Figure 5—figure supplement 1*). The

immediate structural consequences are: (1) an upward shift of the constriction point, which moves from L351 or W358 (upon reconstruction of the unresolved side chains in 4DW1) to A347 and (2) a significant kinking of the TM2 helices at residues V354-I355. Most importantly, the structural reorganization of the TMD occurs while keeping the ion pore open (*Figure 5C*). In this respect, it is important to stress that in all simulations of the open state carried out in the absence of MAM, the ion pore systematically shut under the effect of thermal fluctuations (*Figure 5C*; see 'Materials and methods').

Finally, the analysis of the end-to-end distance distributions of MAM sampled by the simulations of vertically and horizontally fused receptors provides additional information on the nature of the cross-linking experiments reported here. The comparison of the distributions of fused *versus* free MAM (*Figure 1—figure supplement 1C*) makes it clear that: (1) cross-linking reduces the configurational freedom of the photo-switcher quite significantly; (2) the end-to-end distance distributions of fused MAM overlap significantly with those of free MAM; (3) there is a significant shift of the *cis versus trans* distributions when the photo-linker is bound. Overall, these observations indicate that our cross-linking experiments provide evidence of a well-defined structural change of the TM2 helices during activation. Additionally, the comparison of the end-to-end distributions of free MAM and fused MAM indicates that cross-linking is 'soft', such that the structural stability of the open-channel state must be an intrinsic property of the protein; that is, it is not a rare open state stabilized by MAM-cross-linking, but rather the native active state which has been elicited by photoswitching.

## Bending of TM2 at a critical 'hinge' is essential for activation

Finally, our model of the open state trapped vertically by MAM predicted near its attachment site a striking kink in the middle of TM2 (*Figure 5B*). Although a similar kink has been previously described in X-ray structures (*Hattori and Gouaux, 2012*) in proximity of a highly conserved glycine residue (zfP2X4 G350; rP2X2 G342), its functional role in channel opening remains unclear (*Li et al., 2004*; *Khakh and North, 2012*). We hypothesized that bending of TM2 helices around this glycine would facilitate the inner and outer ends of adjacent TM2 helices to come closer together in the open state.

To test this assumption, we mutated this residue into proline (G342P), which is known to facilitate bending of α-helices, and found a dramatic 25-fold increase in ATP sensitivity compared to the WT P2X2 (*Figure 6A and B* and *Figure 1—source data 2*). By contrast, a 2.4-fold decrease of ATP sensitivity was observed for the G342A mutant, which is consistent with the fact that alanine residue rigidifies α-helices. These data demonstrate that TM2 helices in the P2X2 receptor contain a critical 'hinge', which facilitates their vertical bending, allowing the inner and outer ends of adjacent TM2 helices to come closer during activation.

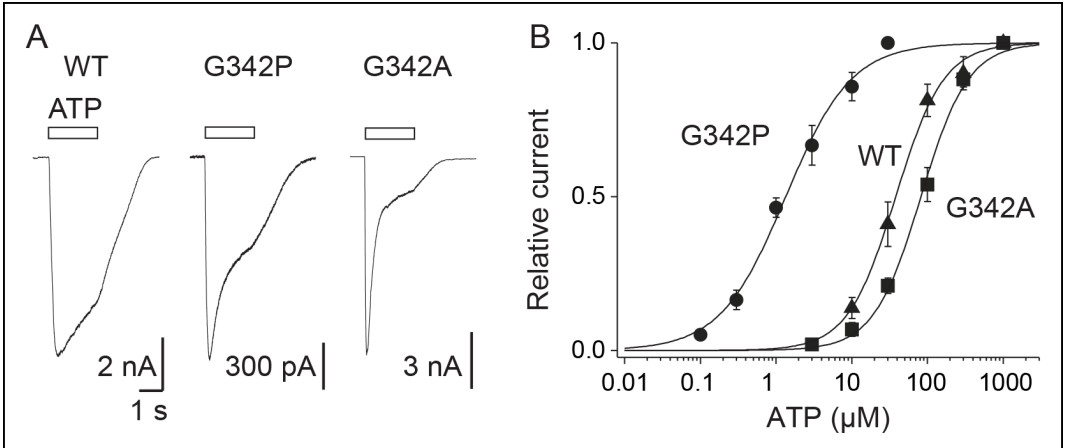

**Figure 6.** Proline mutation supports bending of the TM2 helices during P2X activation. (**A**) Whole-cell currents evoked by ATP (at saturating concentrations) from HEK cells expressing the wild-type (WT) P2X2 receptor (1000 µM), G342P (100 µM) or G324A (1000 µM) mutant. (**B**) Concentration-response relationships for ATP at WT P2X2 receptor and mutant receptors, as indicated (n = 4–6 cells; mean ± s.e.m.). The Hill equation was fit to the normalized data.

## Discussion

In this study, we report on the use of molecular photo-switchable tweezers in conjunction with MD simulations to explore the function of a trimeric ion channel. We show that a vertical decrease of the distance separating the inner and outer ends of adjacent pore-lining TM2 helices, likely through bending at a critical 'gating hinge', is essential for activation of the P2X2 receptor.

Tethered molecular photo-switches have been already employed for the optical control of many receptors and ion channels (*Volgraf et al., 2006*; *Fehrentz et al., 2011*; *Tochitsky et al., 2012*; *Levitz et al., 2013*), including the trimeric acid-sensing ion channels (ASICs) (*Browne et al., 2014*) and ATP-gated P2X receptors (*Lemoine et al., 2013*; *Browne et al., 2014*), mainly to achieve an optogenetic mimicry of neuronal signaling. They were also developed as biophysical tools to control conformational changes of peptides, proteins, or other biomolecules (*Beharry and Woolley, 2011*; *Szymanski et al., 2013*), but probing ion-channel activation or desensitization processes using such photo-switches still remains very limited (*Browne et al., 2014*; *Reiner and Isacoff, 2014*). Therefore, the photo-switchable tweezers strategy presented here is distinct from the optogating approach described previously (*Lemoine et al., 2013*), and provides distance restraints related to gating at physiological conditions. Introducing them as input in MD studies of P2X receptors allowed us to stabilize the open-channel state for the first time in simulation. Interestingly, the resulting model differs substantially from the X-ray structure of the ATP-bound state of zfP2X4, particularly in the TMD.

There are at least three major issues that may arise from the photo-switchable tweezers strategy. The first one is related to the flexibility of the photo-linker (*Beharry and Woolley, 2011*), as we showed that free MAM in solution is not completely rigid, in particular the *cis* isomer, whose end-to-end distance distribution spans from 4 to 22 Å. However, when cross-linked to the TMD of the receptor, we provide evidence that the end-to-end distance distribution shrinks considerably due to the steric constraints imposed by the surrounding protein and membrane. Thus, cross-linked MAM can be considered as a photo-switchable molecular ruler. The second point is the physiological relevance of the light-gated motions. We show here that light-gated channels exhibit biophysical features of the WT receptor activated by ATP, suggesting that the discrete conformational states elicited by light closely resemble those populated by ATP. Most importantly, we found that light-induced motions of the TMD positively modulate ATP function, thus indicating an efficient crosstalk between the ion pore and the distant neurotransmitter-binding sites, a mechanism that is reminiscent of the action of positive allosteric modulators. It remains to be determined, however, whether or not pathways elicited by light that lead to channel opening and closing are similar to those triggered by ATP. The third potential shortcoming is the uncertainties of the extent of 'mixed channels' formed from a heterogeneous population of cross-linked subunits. Although we cannot formerly rule out this possibility, experiments with concatemers rather suggest that mixed channels, if present, occur insignificantly. In particular, the experiments using the OC/CO/CC construct that, in principle, allows cross-linking of either two (vertical) MAM molecules between I328C and S345C or only one (horizontal) MAM molecule between two adjacent I328C subunits, but does not allow both horizontal and vertical cross-linking reactions within the same channel, supports the hypothesis that the currents induced by UV light originate mostly from a homogeneous population bearing two vertical MAM cross-linkers (*Figure 3—figure supplement 3*). In addition it should be stressed that intrinsic formation of disulfide bridges may restrict the extent of cross-linking, but the formation of such disulfides was rather low in our conditions. Our approach thus offers a simple and unique strategy to probe directly in physiological conditions the large-scale functional motions of membrane-embedded proteins.

The first main finding of this study is a set of interatomic distances probed within the TMD both in the resting and the active state of the receptor (*Figure 5A and B*). Our data confirm the lateral expansion of the outer ends of the TM helices (*Li et al., 2008*; *2010*), which was observed in the X-ray structure of the ATP-bound state (*Hattori and Gouaux, 2012*), and functionally probed recently by the shorter photo-linker 4,4'-bis(maleimido)azobenzene (BMA) (*Browne et al., 2014*). However, our findings provide evidence that the simulated end-to-end distance of *trans*-MAM when cross-linked horizontally (~25 Å) is substantially shorter than the distance separating the β-carbons of two adjacent I328 residues in the X-ray structure of the ATP-bound state (~28 Å). The same holds true for the vertical distance probed by *cis*-MAM between the β-carbons of I328 and S345 from adjacent subunits, which appears as overstretched in the X-ray structure of the open channel. Importantly, no

light-gated currents were recorded at P329C, a mutant for which the distance separating the β-carbons of two adjacent P329 in the X-ray structure of the ATP-bound state (~23 Å) was expected to match the end-to-end distance of *trans*-MAM, while the shorter photo-linker BMA was recently shown to optically control the channel activity of this mutant (*Browne et al., 2014*). Incorporating all these structural restraints in the new model of the active state provides compelling evidence that the extent of the outward expansion is reduced by ~3 Å relative to the ATP-bound X-ray structure (*Hattori and Gouaux, 2012*). As a direct consequence, the large inter-subunit crevices visualized by the crystals in the middle of the TMD disappear, and the TMD becomes more tightly packed. The existence of these crevices represents a matter of controversy and recent modeling supported by experimental bridging data has suggested that lipids would be able to cross these gaps and block ion conduction in the bound ATP state (*Heymann et al., 2013*). Our data provide evidence that these crevices are likely artifacts of crystal environment, although we do not rule out the possibility that lipid binding to these interfaces may modulate receptor activity (*Rothwell et al., 2014*).

The second main finding of this study is that the inner and outer ends of adjacent TM2 helices come closer in the open state, most likely through the vertical bending of these helices at a critical gating hinge. This conclusion is supported by the relaxation of the ATP-bound state of zfP2X4 by explicit water/membrane MD with fused MAM, which produced a significant reorganization of the TMD with an open pore stabilized by a new TM2-TM2 interface formed by L351 (V343 in rP2X2) of one subunit and G343 and A347 (A335 and T339 in rP2X2, respectively) of another. The simulations suggest that the new interface is produced by a conformational change of the extracellular end of the pore-lining helices TM2, which straighten relative to the X-ray structure, pack closer together around the region of L351 and lock the channel in a stable, open-pore state. The immediate structural consequences are: i) an upward shift of the constriction point to A347 that is equivalent to T339 in rP2X2, a residue shown to be critically involved in channel gating (*Cao et al., 2007*), and, ii) a significant kinking of the TM2 helices at residues V354-I355. Consistent with this observation, we found that mutating the nearby residue G350 (G342 in rP2X2) into proline, which is expected to promote α-helical bending, dramatically increases the ATP sensitivity in rP2X2. Residue G342 is highly conserved in the P2X family and was shown to be critically involved in ion pore dilation (*Khakh et al., 1999*). A recent work has reported that a missense mutation at this position in the human P2X2 receptor is associated with hereditary hearing loss, suggesting a vital role for this particular conserved glycine residue (*Faletra et al., 2014*). Additionally, G342 is located in proximity to G344, another critical residue for gating (*Fujiwara et al., 2009*). Our data thus demonstrate that during activation the TM2 helices bend through a critical 'gating hinge', a mechanism that is reminiscent of the conformational change involved in K$^+$ channels gating (*del Camino and Yellen, 2001*; *Jiang et al., 2002*).

The emerging model of the open state trapped vertically by MAM is generally consistent with previous experimental and modeling work (*Kracun et al., 2010*; *Li et al., 2010*; *Heymann et al., 2013*). A closer comparison with the model of *Heymann et al. (2013)* shows, however, significant changes in the tertiary structure of the TM2 helices as well as the quaternary organization of the TMD. In our open-state model, the upper region of the TM2 helices is straight and the open-pore conformation is stabilized by the new TM2-TM2 interface. These structural features are different to those of the model of Heymann et al., which displays no tertiary change of the helices compared to the ATP-bound X-ray structure and shows a distinct TM2/TM2 interface that is located below the characteristic kink at residue G350, although some of these interfacial residues (L351 and A347) are common with those of the present model. In addition, the new model of the open state appears to be inconsistent with the Cd$^{2+}$ bridging experiment at position 343 in rP2X2 (L351 in zfP2X4) (*Li et al., 2010*). In fact, the distances between the Cβ atoms of the L351 residues, which correspond to Cd$^{2+}$ coordinating residues (V343C) in rP2X2 (*Li et al., 2010*), are too far apart (11.6 Å) to make a successful metal-coordination site in our model of the open state, even in alternative rotameric states where the Cγ atoms of L351 lie approximately 9 Å apart (full coordination occurs when this distance falls to approximately 6 Å). This said, the side chains of L351 are facing the ion pore lumen in proximity to one another, and it could well be that, when mutated into cysteines, they form an inter-subunit Cd$^{2+}$ coordination site through a major rearrangement of the TM2 helices, perhaps driven by electrostatic interactions with the metal ion. Even though the experimental data support the hypothesis that Cd$^{2+}$ binding occurs in the open state of the receptor (*Li et al., 2010*), this constraint was not included in our model because it would cause sufficient narrowing of the internal

pore that would be incompatible with an open-conducting state. Whether or not the disagreement between our new model of the open state and the inter-subunit $Cd^{2+}$ binding site would be sufficient to disprove its structural details is questionable and further (crosslinking) experiments are required.

Depending on the cross-linking location, MAM is able to trap (at least) two distinct open-channel conformations, as evidenced by differences in desensitization kinetics and NMDG permeability. Although NMDG permeability was insignificant in one case (horizontal cross-linking) and partial in the other (vertical cross-linking), our data support the idea that the P2X pore fluctuates between distinct open conformations, and that these photo-switchable tweezers may be useful to investigate the controversial molecular mechanism of pore dilation.

Finally, we found that the vertical rearrangement of two TM2-TM2 interfaces is sufficient to drive channel gating, which is fully consistent with the recent finding that binding of two ATP molecules is sufficient to open the channel (*Stelmashenko et al., 2012*; *Keceli and Kubo, 2014*). However, the situation is different for the lateral separation of the outer ends of TM2 helices, for which we (present study) and others (*Browne et al., 2014*) have shown that breaking of one TM2-TM2 interface is sufficient for gating. As there is no evidence that this movement alters the sensitivity for ATP, the light-induced outward expansion of a single TM2-TM2 interface is likely a local rearrangement of the TMD.

In conclusion, this study provides the first application of photo-switchable derivatives to investigate the mechanism of pore gating in P2X receptors. By incorporating photo-switchable tweezers at engineered sites at the outer ends (horizontal cross-linking) or between the inner and outer ends (vertical cross-linking) of adjacent TM2 helices, we demonstrate that these tools can be used as molecular rulers to probe structural changes involved in activation. The versatility of the strategy makes it a promising approach for dissecting the allosteric mechanisms of other membrane proteins.

## Materials and methods

### Chemical synthesis

All chemicals were purchased from Sigma-Aldrich, Acros Organics or Alfa Aesar in analytical grade. An Agilent MM-ESI-ACI-SQ MSD 1200 SL spectrometer or an Agilent LC-MS Agilent RRLC 1200SL/ESI QTof 6520 was used for ESI analysis. $^1H$ NMR and $^{13}C$ NMR were run at 400 and 100 MHz, respectively. Coupling constants ($J$) are quoted in Hz and chemical shifts ($\delta$) are given in parts per million (ppm) using the residue solvent peaks as reference relative to TMS.

(E)-N,N'-(diazene-1,2-diylbis(4,1-phenylene))bis(2-(2,5-dioxo-2,5-dihydro-1H-pyrrol-1-yl)acetamide) (1)

1-[Bis(dimethylamino)methylene]-1H-1,2,3-triazolo[4,5-b]pyridinium 3-oxid hexafluorophosphate (HATU) (1100 mg, 4.7 mmol) was added to a solution of 2-(2,5-dioxo-2,5-dihydro-1H-pyrrol-1-yl)acetic acid (730 mg, 4.7 mmol) (obtained as described [*Allen et al., 2010*]), 4-[(E)-2-(4-aminophenyl)diazen-1-yl]aniline (200 mg, 0.942 mmol) and diisopropylethylamine (0.800 ml, 4.7 mmol) in 10 ml of acetonitrile/DMF: 2/1. The reaction was carried out for 20 hr under argon atmosphere at room temperature. The solution was concentrated in vacuo. The crude product was dissolved in a small volume of DMF and the product was precipitated using acetone. The precipitate was recovered by filtration, leading to the desired orange solid (252 mg, 0.518 mmol, 55%). $^1H$ NMR (400 MHz, DMSO-$d_6$): $\delta$ = 10.64 (s, 2H), 7.85 (d, 4H, $J$ = 8.7 Hz), 7.75 (d, 4H, $J$ = 8.7 Hz), 7.16 (s, 4H), 4.33 (s, 4H); $^{13}C$ NMR (100 MHz, DMSO-$d_6$): $\delta$ = 170.62, 165.31, 148.83, 141.10, 134.95, 123.46, 119.42, 40.43 (*Figure 1—figure supplement 4*); MS (ESI) (m/z): [M+H]$^+$ calcd. for $C_{24}H_{19}N_6O_6^+$ 487.1361, found, 487.1350.

(E)-N-(4-((4-aminophenyl)diazenyl)phenyl)-2-(2,5-dioxopyrrolidin-1-yl)acetamide (2)

N,N,N',N'-Tetramethyl-O-(1H-benzotriazol-1-yl)uronium hexafluorophosphate (HBTU) (429 mg, 1.13 mmol) was added to a solution of 2-(2,5-dioxopyrrolidin-1-yl)acetic acid (177 mg, 1.13 mmol) (obtained as described [*Bansode et al., 2009*]), 4-[(E)-2-(4-aminophenyl)diazen-1-yl]aniline (200 mg, 0.942 mmol) and diisopropylethylamine (0.192 ml, 1.13 mmol) in 10 ml of acetonitrile/DMF: 2/1. The reaction was carried out for 20 hr under argon atmosphere at room temperature. The solution was concentrated *in vacuo*, dissolved in AcOEt (150 ml), quenched with 150 ml of saturated NaHCO$_3$aq., then extracted with AcOEt (2 × 150 ml). The crude product was purified by column chromatography

on silica (Heptane / EtOAc: 1/1 to 2/8 in vol.), resulting in the desired orange solid (221 mg, 0.63 mmol, 67%). $^1$H NMR (400 MHz, DMSO-d$_6$): $\delta$ 10.47 (s, 1H), 7.73 (d, 2H, $J$ = 8.8 Hz), 7.68 (d, 2H, $J$ = 8.8 Hz), 7.62 (d, 2H, $J$ = 8.8 Hz), 6.65 (d, 2H, $J$ = 8.8 Hz), 6.03 (s, 2H), 4.24 (s, 2H), 2.75 (s, 4H); $^{13}$C NMR (100 MHz, DMSO-d$_6$): $\delta$ 177.14, 165.53, 152.47, 148.29, 142.82, 139.59, 124.82, 122.50, 119.41, 113.36, 41.12, 28.06 (*Figure 1—figure supplement 5*); MS (ESI) (m/z): [M+H]$^+$ calcd. for C$_{18}$H$_{18}$N$_5$O$_3$$^+$352.1410, found, 352.1408.

(*E*)-2-(2,5-dioxo-2,5-dihydro-1*H*-pyrrol-1-yl)-*N*-(4-((4-(2-(2,5-dioxopyrrolidin-1-yl)acetamido)phenyl) diazenyl)phenyl)acetamide (3)

1-[Bis(dimethylamino)methylene]-1H-1,2,3-triazolo[4,5-b]pyridinium 3-oxid hexafluorophosphate (HATU) (380 mg, 0.62 mmol) was added to a solution of 2-(2,5-dioxopyrrolidin-1-yl)acetic acid (96 mg, 0.62 mmol) (obtained as described [*Allen et al., 2010*]), (*E*)-*N*-(4-((4-aminophenyl)diazenyl)phe-nyl)-2-(2,5-dioxopyrrolidin-1-yl)acetamide (180 mg, 0.52 mmol) and diisopropylethylamine (0.105 ml, 0.62 mmol) in 10 ml of acetonitrile/DMF: 2/1. The reaction was carried out for 19 hr under argon atmosphere at room temperature. The solution was concentrated *in vacuo*, dissolved in AcOEt (150 ml), quenched with 150 ml of saturated NaHCO$_3$aq., then extracted with AcOEt (2 × 150 ml). The crude product was dissolved in a small volume of DMF, and the product was precipitated using ace-tone. The precipitate was recovered by filtration, leading to the desired orange solid (486 mg, 0.31 mmol, 60%). $^1$H NMR (400 MHz, DMSO-d$_6$): $\delta$ = 10.62 (s, 1H), 10.59 (s, 1H), 7.86 (d, 4H, $J$ = 8,7 Hz), 7.75 (d, 4H, $J$ = 8,7 Hz), 7.16 (s, 4H), 4.33 (s, 2H), 4.26 (s, 2H), 2.76 (s, 4H); $^{13}$C NMR (100 MHz, DMSO-d$_6$): $\delta$ = 177.80, 171.04, 165.75, 165.49, 148.24, 141.32, 135.26, 123.85, 119.91, 41.46, 40.79, 28.41 (*Figure 1—figure supplement 6*); MS (ESI) (m/z): [M+H]$^+$ calcd. for C$_{24}$H$_{21}$N$_6$O$_6$$^+$ 489.1, found, 489.0.

MAM and MAS were dissolved in dimethyl sulfoxide (DMSO, Euromedex) to make stock solutions (0.1–5 mM) and diluted in standard extracellular solution to 1–50 µM for labeling.

## Molecular biology

Site-directed mutations were introduced into the rat P2X2 cDNA in the pcDNA3.1(+) using KAPA HiFi HotStart PCR kit (Cliniscience, France) as described previously (*Jiang et al., 2010*). All mutations were confirmed by DNA sequencing.

The WT trimeric P2X2 concatemer was obtained in three steps as described (*Browne et al., 2011*) and ligated into pcDNA3.1(+) vector. A C-terminal EE-epitope tag was also included in each subunit. To introduce mutations into the concatemer, each sequence encoding a monomer was cut from the concatemer at the according restriction sites and ligated into a shuttle vector. Site-directed mutagenesis was performed on the cDNA encoding individual subunit using KAPA HiFi HotStart PCR kit (Cliniscience, France), and confirmed by DNA sequencing. This product was then ligated back into the concatemer, and correct insertion was controlled by restriction enzyme digestion. The trimeric P2X2-3T concatemer was first obtained (OO/OO/OO), in which the three native cysteines (C9, C348 and C430) of each subunit were mutated to threonine (*Li et al., 2008*). Cysteine mutations at I328 and/or S345 were then introduced into this concatenated P2X2-3T background as indicated in *Figure 3—figure supplement 3*.

## Gene expression in cultured cells

HEK-293 and TSA-201 cells were cultured and transiently transfected with the pcDNA3.1(+) vectors (0.05–2 µg) and a green fluorescent protein cDNA construct (0.3 µg) as previously described (*Jiang et al., 2010*).

## Biochemistry

Cell-surface expression of concatemers was determined using the thiol-cleavable, membrane-imper-meant reagent sulfosuccinimidyl-2-(biotinamido)ethyl-1,3-dithiopropionate (Sulfo-NHS-SS-Biotin, ThermoFisher Scientific, France). TSA-201 cells were transfected with the pcDNA3.1(+) vectors con-taining the concatenated constructs (5 µg) and used 1 day after. Cells were solubilized in lysis buffer and the supernatant was incubated overnight with neutravidin-agarose beads (ThermoFisher Scientific, France) as previously described (*Jiang et al., 2010*). Protein samples were run on a 4–15% SDS–PAGE in Tris/Glycine/SDS running buffer (Bio-Rad, France). Samples were transferred to a nitro-cellulose membrane as described (*Jiang et al., 2010*), which was then incubated in TPBS (PBS

supplemented with 1% nonfat dry milk, 0.5% bovine serum albumin, and 0.05% Tween 20) containing a peroxidase-conjugated Glu-Glu Tag Rabbit Polyclonal antibody (EYMPME) for 2 hr (dilution 1:5000) at room temperature. Blots were developed with the Amersham ECL Prime Western blotting detection reagent (Dominique Dutscher, France).

Cross-linking of cell-surface receptors was performed as follows. Adherent intact TSA-201 cells in dishes were incubated for 20 min under gentle agitation with 30 µM MAM in the presence of 3 µM ATP in ice-cold PBS containing 154 mM NaCl, 2.68 mM KCl, 4.2 mM $Na_2HPO_4$, 1.47 mM $KH_2PO_4$, pH 7.0, supplemented with 1 mM $MgCl_2$ and 0.4 mM $CaCl_2$. Dishes were rapidly washed with PBS and incubated with Sulfo-NHS-SS-Biotin in PBS pH 8.0 as described above. The nitrocellulose membrane was incubated in TPBS buffer overnight at 4°C with mouse anti-c-Myc antibody (ThermoFisher Scientific, France) diluted at 1:2500. After three washes with TPBS, the blot was incubated with peroxidase-conjugated sheep anti-mouse antibody for 2 hr (dilution 1:10,000; GE Healthcare life Sciences, France) at room temperature and further washed three times with TPBS, and developed with the Amersham ECL Prime Western blotting detection reagent (Dominique Dutscher, France).

## Patch-clamp electrophysiology

Whole-cell recordings were performed 24–48 hr after transfection. Before recordings, cells were incubated for 20 min with MAM (unless stated otherwise) or for 40 min with MAS in standard extracellular solution (50 µM MAM for the horizontal screening, 1–15 µM MAM for the vertical screening, 1–15 µM MAS) and 3 µM ATP (to increase the accessibility of residues) in the dark at room temperature. After treatment, cells were extensively washed out. Patch pipettes (3–5 MΩ) contained 140 mM KCl, 5 mM $MgCl_2$, 5 mM EGTA, 10 mM Hepes, adjusted to pH 7.3 with NaOH. The standard extracellular solution contained 140 mM NaCl, 2.8 mM KCl, 2 mM $CaCl_2$, 2 mM $MgCl_2$, 10 mM glucose, 10 mM Hepes, adjusted to pH 7.3 with NaOH. All solutions were maintained approximately at 300 mOsm. Cells were voltage-clamped to -60 mV using the EPC10 (HEKA) amplifier, and data were recorded with PATCHMASTER software.

For relative permeability measurements, an agar bridge containing 3 M KCl connected the bath and indifferent electrode. The intracellular solution comprised 147 mM NaCl, 10 mM EGTA, 10 mM Hepes, adjusted to pH 7.3 with NaOH, or 140 mM CsCl, 5 mM MgCl2, 5 mM EGTA, 10 mM Hepes, adjusted to pH7.3 with CsOH. The standard extracellular solution was changed to symmetrical NaCl external solution and a voltage ramp pulse (from −120 to 80 mV; 165 ms duration) was applied. The solution was then exchanged with one of the following solutions, and another voltage ramp was applied: mannitol (Man) solution, sodium isethionate solution (Na-Ise), $CaCl_2$ (Ca) solution and NMDG solution. Composition of these solutions was described elsewhere (*Lemoine et al., 2013*) except for glucose, which was 10 mM. Voltage ramps were applied 300 ms after light switching. For light-gated currents, we calculated reversed potential ($E_{rev}$, mV) from voltage ramps after subtracting photocurrents to those recorded in the dark.

Single-channel recordings using outside-out configuration were carried out using HEK-293 cells at room temperature 24 hr after transfection. Recording pipettes pulled from borosilicate glass (Harvard Apparatus) were coated with Sylgard 184 (Dow Corning Co.) and fire polished to yield resistances of 6–20 MΩ. The holding potential was -120 mV. The extracellular solution contained 147 mM NaCl, 2 mM KCl, 1 mM $CaCl_2$, 1 mM $MgCl_2$, 10 mM Hepes, and 13 mM glucose, pH 7.3. The intracellular solution contained 147 mM NaF, 10 mM Hepes, and 10 mM EGTA, pH 7.3. Data were sampled at 4–10 kHz, and low-pass filtered at 2.9 kHz. For off-line analysis, data were refiltered to give a cascaded filter cutoff frequency of 1–2 kHz (*Jiang et al., 2011*).

Illumination of cells was achieved as described previously (*Lemoine et al., 2013*) with LEDs directly coupled to the microscope. The measured output intensities for wavelengths for 365 and 525 nm were 8.1 and 4.1 mW/mm$^2$, respectively. Drug applications were carried out as described previously (*Jiang et al., 2010*).

## Molecular modeling

Models of the P2X trimers with or without attached MAM photo-switchable tweezers were produced starting from the crystal structures of the zfP2X4 receptor in the apo, closed-channel (PDB ID: 4DW0) and ATP-bound, open-channel (4DW1) states (*Hattori and Gouaux, 2012*). Unresolved side

chains were modeled in CHARMM (*Brooks et al., 2009*) as well as the missing terminal residues (modeled as α-helical extensions of TM1 and TM2) corresponding to the 28–365 residue range of the experimental truncated zfP2X4 construct (*Hattori and Gouaux, 2012*). The protonation state of the ionizable residues was accessed by $pK_a$ calculations following the multiple-site titration approach based on continuum electrostatics (*Bashford and Karplus, 1990*; *Bashford and Karplus, 1991*). The results obtained using APBS (*Baker et al., 2001*) and Karlsberg (*Rabenstein and Knapp, 2001*) indicated that all residues are in their standard protonation state at pH 7, independently of the ATP agonist. Cysteine residues were introduced at positions I336 (rP2X2: I328) or I336/N353 (S345) to incorporate the MAM molecule by fusing the reactive maleimide moieties with the thiol group of the engineered cysteines. All R/R, R/S, S/R, and S/S stereoisomers of the protein-fused MAM were treated simultaneously using four non-interacting copies (*Roitberg and Elber, 1991*) of the photo-switchable cross-linker. Either one horizontal (I336C/I336C) or three vertical (I336C/N353C) MAM molecules were incorporated in both the closed and the open-channel models. The protein trimers were then embedded in a pre-equilibrated 1-palmitoyl-2-oleoyl-*sn*-phosphatidylcholine (POPC) bilayer and fully solvated (including the open pore) with modified TIP3P (*Durell et al., 1994*) water molecules using VMD (*Humphrey et al., 1996*). The net charge of the system was neutralized by adding a 150 mM equivalent of sodium and chloride ions. The resulting all-atom constructs (~138000 atoms each) were modeled using the CHARMM force-field version 36 (*Best et al., 2012*) with MAM parameters obtained from the CHARMM general force-field (*Vanommeslaeghe et al., 2010*).

The molecular systems were then subjected to energy minimization (5000 steps) followed by short MD equilibrations with periodic boundary conditions and Particle Mesh Ewald long-range electrostatics. Harmonic restraints were set on the positions of the phosphorus and adenosine heavy atoms (1 kcal/mol/Å$^2$) of the ATP agonists and of the Cα and Cβ atoms (1.0 and 0.5 kcal/mol/Å$^2$, respectively) of the protein except for the TM2 helices, the cross-linked Cys residues, and the added terminal residues. Following a short thermalization (600 ps), a 2-ns MD relaxation was performed in the NPT ensemble at 310 K and 1 atm with vanishing positional restraints. Because in the absence of MAM the open-state conformation captured in the crystal was unstable in simulation, that is, spontaneous shut of the ion pore within a few (<5) nanoseconds of equilibration independently of the initial setups (i.e. equilibration procedure, protonation state of residues, membrane composition, etc.), for the simulations of the open state, the TM1-TM2 interface was stabilized by incorporating structural information obtained from recent intra-subunit Cd$^{2+}$ bridging essays (*Heymann et al., 2013*). Specifically, the trigonal-planar Cd$^{2+}$ coordination between the side chains of N35 (Cγ), N353C (Sγ), and C356 (Sγ) was mimicked by introducing harmonic distance restraints (reference distance 4.2 Å, force constant 1 kcal/mol/Å$^2$). Structural information related to the inter-subunit Cd$^{2+}$ coordination site of*Li et al., (2010)*, which involves the side chains of L351 (rP2X2: V343), was not included in the modeling.

All MD simulations of P2X were performed using NAMD (*Phillips et al., 2005*) version 2.10. Six independent 50 ns-long unrestrained MD simulations of the free MAM molecule in solution (~10,000 atoms) were performed using ACEMD (*Harvey et al., 2009*) in the NVT ensemble at 310 K for the *cis* and *trans* configurations and for the R/R, S/R, and S/S stereoisomers. Trajectories were analyzed using VMD and Wordom (*Seeber et al., 2007*), and molecular snapshots were rendered using PyMOL (Schrödinger, LLC). Profiles of the TM pore radius along the axis were calculated using the program HOLE (*Smart et al., 1996*) for backbone atoms only to emphasize global TM2 displacements. The mean distances between the S–S atoms (free MAM) obtained along the MD simulations were computed by averaging over the R/R and S/S stereoisomers (25,000 snapshots each) and twice the R/S (resp. S/R) stereoisomer (2 × 25,000 snapshots) for a total of n = 100000 for both the *cis* and *trans* configurations. The mean distances between the Cβ–Cβ of the cross-linked Cys residues were obtained for all four stereoisomers per MAM over the 200 snapshots saved along the 2 ns-long MD simulation, that is, n = 800 or 2400 for one or three MAM molecule(s), respectively. The normalized probability distributions of the S–S or Cβ–Cβ distances were obtained by clustering all distance values using a bin width of 0.2 Å. For the free MAM simulations, the peak probability was then selected as the mean of the distribution, and the standard deviation was obtained after performing a least-square fit with a normal distribution over the full dataset of probabilities using a modified Levenberg-Marquardt algorithm (*Kelley, 1999*). For the simulations with MAM fused to the protein, both the mean and the standard deviation were obtained from the gaussian least-square fit.

## Data analysis

For data analyses, FitMaster (HEKA Electroniks, v2x69) and IGOR PRO (WaveMetrics, v6.32A) software were used. Experiments were repeated several times after, at least, two independent transfections. Data from dose–response relationships were fit to the Hill equation as described previously (*Jiang et al., 2010*). For determining ATP dose–response relationship for the I328C mutant at visible light, we subtracted $I_{(ATP+light)}$ from $I_{(light)}$, where $I_{(ATP+light)}$ is the maximal current evoked by ATP (at a given concentration) during light irradiation, $I_{(light)}$ is the maximal light-gated current recorded in the absence of ATP.

Increase of ATP maximal current was determined by the following equation: $(I_{(ATP+light)} - I_{(light)}) / I_{(ATP)}$, where $I_{(ATP+light)}$ is the maximal current evoked by a saturating concentration of ATP during light irradiation, and $I_{(ATP)}$ is the maximal current evoked by a saturating concentration of ATP in the dark. Apparent ATP desensitization of the I328C/S345C mutant was determined by calculating the ratio of the remaining current recorded just before the end of the ATP application and the peak current during the same application.

Fitting procedures to access the time constant were based on the single-exponential decay equation function: $I_t = I_0 + A \exp(-t/\tau)$, where $I_0$ and A are the residual current and maximal amplitude, respectively, t is the time in seconds, and $\tau$ is the time constant in seconds.

Channel events were detected by using TAC software (Bruxton Co.) and conductance levels were measured by all-points amplitude histograms fit to Gaussian distributions. For relative permeability measurements, values of $E_{rev}$ were corrected for liquid junction potentials, which were calculated using IGOR PRO, and used to determine the permeability ratios ($P_X/P_{cations}$) as described previously (*Lemoine et al., 2013*).

## Acknowledgements

The authors thank Pr. M Goeldner and K Dunning for critical reading of the manuscript. This work was supported by the Centre National de la Recherche Scientifique, the International Center for Frontier Research in Chemistry (icFRC) and grants from the Agence Nationale de la Recherche (ANR-11-BSV5-001-01 and ANR-11-BSV5-001-02). This work was granted access to the HPC resources of CCRT under the allocation 2014-[076644] made by GENCI (Grand Equipement National de Calcul Intensif). We acknowledge the Partnership for Advanced Computing in Europe (PRACE) for awarding us access to resources at the SuperMUC center in Leibniz (Germany). HPC resources from the Mesocentre at the University of Strasbourg are also acknowledged. CH was supported by a doctoral fellowship from the icFRC (CONV/2011/01/CIRFR-N°157). DL was supported by a postdoctoral fellowship from the Agence Nationale de la Recherche (ANR-11-BSV5-001-01). NC received support from IdEx Projet Interdisciplinaire (Grant No. RIDEX04).

## Additional information

### Funding

| Funder | Grant reference number | Author |
|---|---|---|
| Centre International de Recherches aux Frontières de la Chimie | CONV/2011/01/CIRFR-No. 157 | Chloé Habermacher Thomas Grutter |
| Agence Nationale de la Recherche | ANR-11-BSV5-001 | Damien Lemoine Alexandre Specht Thomas Grutter |
| Université de Strasbourg | RIDEX04 | Nicolas Calimet Marco Cecchini Thomas Grutter |

The funders had no role in study design, data collection and interpretation, or the decision to submit the work for publication.

**Author contributions**

CH, NC, AS, Conception and design, Acquisition of data, Analysis and interpretation of data; AM, DL, LP, Acquisition of data, Analysis and interpretation of data; MC, TG, Conception and design, Analysis and interpretation of data, Drafting or revising the article

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
