## [Decision Letter]

Thank you for submitting your work entitled "Photo-switchable tweezers illuminate pore-opening motions of an ATP-gated P2X ion channel" for peer review at *eLife*. Your submission has been favorably evaluated by Richard Aldrich (Senior Editor), a Reviewing Editor, and two reviewers, one of whom has agreed to reveal his identity: Kenton Swartz (peer reviewer #1).

The reviewers have discussed the reviews with one another and the Reviewing Editor has drafted this decision to help you prepare a revised submission.

Two reviewers have submitted comments on your manuscript and we are recommending that the work be published in *eLife*.

The comments require no new experiments, but some modification of the text and extension of the Discussion to cover some limitations of the approach.

Here are the specific suggestions that need to be addressed from Reviewer 1:

1) A related paper (Heymann et al., 2013) is cited as a modeling study that argues for closing the gaps between subunits that are evident in the zfP2X4 X-ray structure. While this is accurate, what is not stated is that paper also reports experimental metal bridging results for residues between TM1 and TM2 that are consistent with the X-ray structure. To be fair, I think it’s important to distinguish which features of the X-ray structure are compatible and which are not, and would suggest modifying the text in paragraph three of the Introduction and paragraph four of the Discussion accordingly. Also, to what extent is the new model consistent with those metal bridges or the model emerging from that work? This should also be added to the Discussion in the subsection “Improved model of the open-channel state”.

2) In the subsection “Vertical cross-linking indicates shortening of the distance separating the inner and outer ends of adjacent TM2 helices during activation”, you sate 'To distinguish vertical versus horizontal cross-linking […]' Shouldn't this consideration of possible mechanisms also consider monovalent MAM reactions since those were seen at I328C using MAS?

3) Concerning the discussion in paragraph six of the Discussion about internal Cd bridges, I would suggest stating what types of structural change would be needed to bring the relevant residues within bridging distance. You should then state that you haven't done this because you think it’s possible that bridge formation causes sufficient narrowing of the internal pore to form a stable Cd site. Indeed, the modeling in Heymann et al. show that quite subtle structural changes are needed to form this metal binding site. Also, is there a reason to cite Kracun et al. in paragraph three of the Introduction, but not in in paragraph six of the Discussion?

4) Paragraph four of the Discussion. Expansion of the external pore is also demonstrated by state-dependent accessibility studies reported in Li et al., 2008 and 2010.

5) It would be helpful for the authors to comment on the relatively low extent of crosslinking evident in Figure 1. Is this because the dimer band transfers less efficiently?

Reviewer 2 had more reservations as follows:

Most problematic are the experiments in which the authors attempt to probe 'vertical motions' between an interfacial residue (I328C) and other positions in the channel transmembrane domains. It is impossible given the current data to know the extent of mixed channels in these experiments (that is those bearing subunits containing I328C-I328C crosslinking versus I328C to some other residue). Further, the biochemistry suggests that for many of the constructs there is some intrinsic formation of disulfides. For example, compare Y47C without crosslinker to D57C and I328C in Figure 1). It is impossible for the authors to know if the crosslinking is complete in their experiments, or to know the proportions of mixed channels versus those having homogeneous crosslinks. Due to this imprecision, one cannot draw clean conclusions about bending or helices or the extent of opening versus what has been proposed from the crystal structures.

Although Reviewer 2 felt these were severe limitations for the conclusions, we think you can address these in the Discussion. In particular, please state the potential shortcomings of formation of disulfides and uncertainties about 'mixed channels" and how this might affect your conclusions.

---

## [Author Response]

*Here are the specific suggestions that need to be addressed from Reviewer 1:*

*1) A related paper (Heymann et al., 2013) is cited as a modeling study that argues for closing the gaps between subunits that are evident in the zfP2X4 X-ray structure. While this is accurate, what is not stated is that paper also reports experimental metal bridging results for residues between TM1 and TM2 that are consistent with the X-ray structure. To be fair, I think it’s important to distinguish which features of the X-ray structure are compatible and which are not, and would suggest modifying the text in paragraph three of the Introduction and paragraph four of the Discussion accordingly. Also, to what extent is the new model consistent with those metal bridges or the model emerging from that work? This should also be added to the Discussion in the subsection “Improved model of the open-channel state”.*

We agree on these comments. As suggested by the referee, we modified the text in the Introduction to clarify which features of the X-ray structures are compatible with functional and modeling data and which are not. The text can be read as follows: ‘The mechanism of gating based on the crystal structures is largely consistent with previous functional and modeling data obtained on the ECD (Jiang et al., 2010; Du et al., 2012; Jiang et al., 2012; Lorinczi et al., 2012; Roberts et al., 2012; Hausmann et al., 2013; Huang et al., 2014; Stelmashenko et al., 2014; Zhao et al., 2014). However, there are areas of discordance between the X-ray structures and the available data at the level of the transmembrane pore. Although the location of the gate (Samways et al., 2014), the relative position and gating motion of TM1 and TM2 within the individual subunits (Li et al., 2008; Heymann et al., 2013) and the movement of the outer ends of the TM helices (Li et al., 2008; Kracun et al., 2010; Li et al., 2010; Heymann et al., 2013; Browne et al., 2014) inferred from experimental data are in qualitative agreement with the crystal structures, there are reasons to question whether the ATP-bound structure provides an accurate blueprint of a native open-channel pore’.

Secondly, we provide in the Discussion a structural comparison of our model with that of Heymann et al. As the atomic coordinates for the model (of Heymann et al., 2013) are not publicly available, a detailed comparison with our model of the open state is difficult and can be done only qualitatively. Both models consistently show a significant shrinking of the subunits interface in the TMD, which results in the disappearance of the large crevices observed in the X-ray structure of the ATP-bound state (Hattori & Gouaux, 2012). However, significant differences between them are apparent both at the level of the tertiary structure of the TM2 helices and the quaternary organization of the TMD. In our open-state model, the upper part of the TM2 helices straightens and stabilizes the open channel by producing a new TM2/TM2 interface that involves the side chain of L351 from one subunit and G343 and A347 from another. These structural features are different from those of the model of Heymann et al., which based on their **Figure 8A** displays no tertiary change of the TM2 helices compared to the X-ray structure and a distinct TM2/TM2 interface, located well below the characteristic kink (residue G350). Importantly, our model of the open state was shown to preserve an open-channel conformation in fully unrestrained MD simulations of dozens of ns, unlike the X-ray structure of the ATP-bound state, which shuts under the effect of thermal fluctuation in a few nanoseconds independently of the initial setup (subsection “Molecular modelling”). Because our model of the open state incorporates distance restraints at the TM1-TM2 interface corresponding to the Cd^2+^ binding experiments of Heymann et al., it is consistent with these metal bridges by construction.

Finally, we also modified the text in the Introduction and Discussion to state that the paper of Heymann et al. also reports on experimental metal bridging results that support the modeling data.

*2) In the subsection “Vertical cross-linking indicates shortening of the distance separating the inner and outer ends of adjacent TM2 helices during activation”, you sate 'To distinguish vertical versus horizontal cross-linking […]' Shouldn't this consideration of possible mechanisms also consider monovalent MAM reactions since those were seen at I328C using MAS?*

We agree with this statement and we thank the referee for highlighting this possibility. Accordingly, we modified the text that can now be read as follows: ‘To distinguish vertical MAM cross-linking from either horizontal cross-linking or MAM reactions involving only one I328C residue (i.e. such as with MAS), light-gated currents were systematically compared with those originating from the I328C mutant alone.’

*3) Concerning the discussion in paragraph six of the Discussion about internal Cd bridges, I would suggest stating what types of structural change would be needed to bring the relevant residues within bridging distance. You should then state that you haven't done this because you think it’s possible that bridge formation causes sufficient narrowing of the internal pore to form a stable Cd site. Indeed, the modeling in Heymann et al. show that quite subtle structural changes are needed to form this metal binding site. Also, is there a reason to cite Kracun et al. in paragraph three of the Introduction, but not in in paragraph six of the Discussion?*

It appears that major structural changes would be required to form a Cd^2+^ binding site from our model. Even though our current model of the open state is generally consistent with that of Heymann et al., it appears that its open conformation is inconsistent with the Cd^2+^ bridging experiment at position 343 in rP2X2 (L351 in zfP2X4). In fact, even in alternative rotameric states, the Cγ atoms of L351 would lie approximately 9 Å apart and would be unable to coordinate Cd^2+^ when mutated into cysteines (this distance should be ~6 Å for full coordination). Thus, major structural changes would be required to form a Cd^2+^ binding site in the lumen of the ion pore, unlike the model of Heymann et al. As suggested by the referee, we haven’t included the data of the inter-subunit Cd^2+^ binding site because this constraint would cause sufficient narrowing of the internal pore that would be incompatible with an open-conducting state. This point is now stated in the Discussion (paragraph six). We also added citation of Kracun et al.

*4) Paragraph four of the Discussion. Expansion of the external pore is also demonstrated by state-dependent accessibility studies reported in Li et al., 2008 and 2010.*

We agree that these papers also demonstrated the expansion of the external pore. Accordingly, we have included these references in the text as suggested in paragraph four of the Discussion.

*5) It would be helpful for the authors to comment on the relatively low extent of crosslinking evident in Figure 1. Is this because the dimer band transfers less efficiently?*

We agree that the extent of cross-linking is rather low (especially for D57C and Y47C), but it should be noted that currents induced by light are also consistently small following MAM treatment. The reason is unclear but it should be emphasize that the kinetics of MAM cross-linking on D57C and Y47C are unknown and it remains possible that the cross-linking reaction was not complete in our conditions. Furthermore, there is a possibility that a fraction of the cell surface mutant receptors might not be reactive to MAM fusion (possibly due to protein misfolding), thus decreasing the cross-linking efficiency. Finally, it remains possible that for unknown reasons the dimer band transfers less efficiently. As suggested, we added in the Results (paragraph three of subsection “Horizontal cross-linking provides a direct measurement of the outward expansion of the TM2 helices during activation”) a comment on this issue to help the reader to understand the data.*Reviewer 2 had more reservations as follows:*

*Most problematic are the experiments in which the authors attempt to probe 'vertical motions' between an interfacial residue (I328C) and other positions in the channel transmembrane domains. It is impossible given the current data to know the extent of mixed channels in these experiments (that is those bearing subunits containing I328C-I328C crosslinking versus I328C to some other residue). Further, the biochemistry suggests that for many of the constructs there is some intrinsic formation of disulfides. For example, compare Y47C without crosslinker to D57C and I328C in Figure 1). It is impossible for the authors to know if the crosslinking is complete in their experiments, or to know the proportions of mixed channels versus those having homogeneous crosslinks. Due to this imprecision, one cannot draw clean conclusions about bending or helices or the extent of opening versus what has been proposed from the crystal structures.*

We agree with the comment that ‘mixed channels’ formed from a heterogeneous population of cross-linked subunits may affect the conclusion when using the present experimental approach. In our case, we cannot formerly rule out this possibility, however, experiments using concatemers suggest that mixed channels, if present, occur insignificantly. In particular, the experiment using the OC/CO/CC construct that, in principle, allows cross-linking of either two (vertical) MAM molecules between I328C and S345C or only one (horizontal) MAM molecule between two adjacent I328C subunits, but does not allow both horizontal and vertical cross-linking reactions within the same channel, supports the hypothesis that the currents induced by UV light originate mostly from a homogeneous population bearing two vertical MAM cross-linkers (Figure 3—figure supplement 3). We are aware of the potential caveats that may arise from the use of concatemers (see Stelmashenko et al., 2012), however, we present evidence that currents recorded after expression of the different constructs results from channels formed by the intact concatenated proteins (no evidence of protein breakdown as detected by SDS PAGE and all tested concatemers responded robustly to ATP). Concerning the intrinsic formation of disulfide bridges, they can effectively affect the cross-linking efficiency of MAM, but the extent of disulfide formation was rather low in our conditions. All these points are now discussed in the third paragraph of the Discussion to help the reader appreciating the potential shortcomings of the photo-switchable tweezers method.